# OpenReview forum: "Reliable Weak-to-Strong Monitoring of LLM Agents"
_ICLR.cc/2026/Conference — ICLR 2026 Oral_

### Official Review · Reviewer_28fS · 2025-10-20

**Soundness:** 4
**Presentation:** 4
**Contribution:** 4
**Rating:** 8
**Confidence:** 4

**Summary:**

This work stress-tests monitoring frameworks in AI control scenarios. The authors conducted extensive experiments to understand which components in AI control evaluation matter the most—for example, whether monitoring awareness or the monitoring framework itself has a greater impact, and many more. Their results show many valuable insights, one of which is that their hybrid monitoring framework remains effective even as the gap between the attacker and the monitor increases.

**Strengths:**

- The authors ask the right questions to study in AI control eval.
- They conducted extensive and detailed experiments.
- They present valuable insights that can be useful for the safety of humanity.

**Weaknesses:**

The current version is already good work. I was thinking about a similar idea a few months ago, and I am glad the authors executed this very well.
- If the authors have time, I'd be interested to see whether all the insights generalize to other coding setups, e.g., bashbench or APPS, etc.

**Questions:**

This is optional and probably will require a significant amount of time, but could you test some of main experiments on one coding setup, ie, BashBench, and see if the insights generalize?
Happy to increase my score to 10 if you indeed have time to do this.

---

> ### Author Response · Authors · 2025-11-18
>
> We thank R4 for the very positive assessment and for emphasizing the importance of the questions we study. We were particularly excited by your suggestion to test generalization, as we agree it is critical for the paper's impact.
>
> ### Concern 1: Test whether insights generalize to other coding setups (e.g., BashBench).
> We accepted your challenge and have added a new **Appendix H: MRT on BashBench**. We performed extensive experiments to reproduce our main results in this new setting:
>
> * **Data Generation:** We generated 100 BashBench trajectories across the full spectrum of agent setups: unaware, aware, lying (with jailbreaking), and monitor-hijacking attackers, plus honest baselines.
> * **Comprehensive Monitoring:** We ran baseline, hierarchical, sequential, and hybrid monitoring with unaware, aware, and omniscient monitors (12 total monitors per attack setting, using Claude 3.7 Sonnet).
> * **Reproducing Key Findings:** We successfully reproduced **Figures 3 and 4** on BashBench (see Appendix H, Figures 14 and 15), confirming that Findings 1-6 from the main paper hold true in this domain.
> * **Scaling Laws:** We generated an additional 100 trajectories with aware **Claude Sonnet 4 and 4.5** attackers against 4 different monitors (ranging from Claude 3.5 Haiku to GPT-4o). We successfully reproduced the **weak-vs-strong scaling curves** on BashBench (Appendix H, Figure 16).
>
> **Conclusion:** The key insights generalize robustly. Specifically, hybrid monitoring matches or outperforms baselines, and weaker monitors with hybrid scaffolding can successfully supervise significantly stronger agents (up to a specific capability gap) in the BashBench environment.
>
> We believe this extensive new evidence fully addresses your question regarding generalization, and we hope this merits the score increase to 10.

---

> > ### Comment · Reviewer_28fS · 2025-11-18
> >
> > Thanks for running this additional experiment on BashBench. I've read through Appendix H. Overall, this work provides valuable insights, especially the finding that the proposed hybrid method remains robust even as the capability gap between the attacker and monitor increases, and this finding is consistent across SHADE-Arena (realistic long trajectory tasks) and BashBench (coding tasks). This is a critical finding that ideally, frontier AI labs should adopt to safeguard misaligned AIs. I think this paper deserves a spotlight, so I have raised the rating to 10.

---

> > > ### Author Response · Authors · 2025-11-25
> > >
> > > Thank you very much for guiding us toward this generalization experiment and for the recommendation for a spotlight. We agree that demonstrating the robustness of hybrid monitoring across different domains (SHADE-Arena, CUA-SHADE, and now BashBench) is critical for future AI control measures. We are glad that Appendix H addressed your questions regarding generalization, and we are grateful for the opportunity to improve our work.

---

### Official Review · Reviewer_2pXk · 2025-10-28

**Soundness:** 3
**Presentation:** 3
**Contribution:** 2
**Rating:** 6
**Confidence:** 4

**Summary:**

This paper considers monitoring LLM-based tool-using agents for harmful behaviors. This paper introduces a benchmark of computer-use tasks with harmful side tasks which the agent can pursue or not. A set of threat models, including various agent and monitor awareness levels and strategies for evading and detecting, respectively, are introduced to simulate various AI control settings. The authors experiment with various settings in order to elucidate which settings are critical for both agent and monitor success. The experiments indicate several key findings including (a) weaker monitors with the right scaffolding can detect misbehavior of stronger models; (b) awareness improves agent's evasion success, but not monitor success; and (c) agent evasion strategies degrade monitor performance.

**Strengths:**

* The paper is well-motivated, thorough, and well written.
* The paper introduces a foundational framework for evaluating different agents, monitoring strategies, and evasion strategies.
* The experimental setup is well thought out and the conclusions are sound with respect to the experimental results.

**Weaknesses:**

* The threat model presented in the paper is somewhat artificial and potentially unrealistic. One of the main assumptions made in the paper is that the agent is situationally aware. It is controversial whether or not this is a credible threat. Additionally, the agent must be prompted to perform different malicious actions. There is a lot of precedence in the AI evaluation space that this is ok, but it's not necessarily reflective of the true risks and capabilities of agents and monitors.
* The experiments only consider one type of scenario. Namely, the AgentDojo style indirect prompt injection side task. It's unclear whether or not these results generalize to other settings.
* All monitoring strategies evaluated use CoT monitoring. It has been shown in several works that this is not the most reliable means of monitoring.
* One of the main conclusions is that weak monitors can effectively monitor strong agents. I think that this has already been established in the safety community as falling under the "verification is easier than generation" phenomenon.

**Questions:**

* How does the agent know that it is being monitored? Is there evidence of situational awareness or is this just a theoretical possibility?
* In what scenario would the monitor not have access to the agent's LLM? Third party auditing scenarios only?
* Is there a way to guide the agents to be malicious or misaligned without prompting? This seems unrealistic.
* Does the difficult of the main or side task affect the performance of the monitor or the agent?

---

> ### Author Response · Authors · 2025-11-18
>
> We thank the reviewer for the constructive feedback. In response, we have added a new benchmark (BashBench), new analysis on task difficulty, and clarified our threat models.
>
> ### 1. Situational Awareness & “No Access” Settings
>
> > “Is there evidence of situational awareness? How does the agent know?”
>
> This is no longer hypothetical. The **Claude 4.5 Sonnet system card** [1] and recent research [2] explicitly document that models often recognize when they are being safety-tested. We model this realistic "evaluation awareness" by explicitly informing the agent via system prompts (Sec 3.2).
>
> > “In what scenario would the monitor not have access to the agent's LLM?”
>
> In **Appendix J**, we explicitly map this to **third-party auditing** and **API-based oversight**, where auditors only see actions (files, browser) and transcripts, not weights or internal states.
>
> ### 2. Threat Model Realism & Prompting
>
> > “Prompted malicious actions... are not necessarily reflective of true risks.”
>
> We agree our setup is a controlled red-teaming environment. We rely on the standard red-teaming paradigm [3] **to keep the underlying model capability fixed.** Fine-tuning would introduce uncontrolled variables (e.g., reasoning degradation or overfitting), making it impossible to isolate the specific effects of evasion strategies on monitor performance.
>
> **Changes:** We added a **“Threat Model Limitations”** subsection (Sec. 5) to:
> * Frame our setup as stress-testing monitoring *infrastructure* rather than modeling intrinsic misalignment.
> * Explicitly scope our results to "weak-to-strong–style monitoring in a controlled MRT setting."
>
> ### 3. Generalization (New Experiments)
>
> > “The experiments only consider one type of scenario.”
>
> To test generalization across modalities, we **replicated our key experiments on BashBench** (coding sabotage) in **Appendix H**. We reproduce our core findings:
> * **Hybrid monitoring dominates:** The hybrid hierarchical-sequential monitor outperforms baselines (**Figure 14**).
> * **Evasion works:** Agent awareness and evasion strategies degrade monitor performance identically to the original setup (**Figure 15**).
> * **Weak monitors succeed:** Weaker monitors can verify stronger agents given correct scaffolding (**Figure 16**).
>
> ### 4. CoT Monitoring Limitations
>
> > “CoT monitoring... is known to be unreliable.”
>
> We agree; our own results quantify this. In **Appendix E.4 (Fig. 10d, 10e)**, we show that performance degrades significantly when the agent hides parts of the CoT.
>
> **Changes:** We revised **Appendix A** to position our hybrid method as a "best-practice baseline," and our CUA-SHADE benchmark as infrastructure to evaluate future non-CoT methods [4].
>
> ### 5. Novelty of "Verification is Easier"
>
> > “I think that this has already been established... as ‘verification is easier than generation’.”
>
> While the *intuition* exists, prior work [5] shows that weak models often *fail* to monitor strong attackers. We demonstrate that the "verification advantage" is **not innate** to agents—it requires specific architectures.
> * **Baselines fail:** Without our specific hybrid scaffolding, weak monitors *cannot* effectively monitor strong agents.
> * **Evasion boundaries:** We quantify exactly how agent evasion strategies degrade verification.
> We revised **Sec. 2** to clarify that we are stress-testing the *architectural requirements* to maintain this advantage under evasion.
>
> ### 6. Task Difficulty Analysis
>
> > “Does the difficulty... affect the performance?”
>
> We added a detailed analysis in **Appendix E**.
> * **Differential Impact (Fig. 10):** Simple monitors (Baseline/Sequential) perform better on *easier* tasks (high task success rates). However, our **Hierarchical and Hybrid monitors significantly outperform on harder tasks**, where simple monitors fail due to context length and cognitive load.
> * **Granular Comparison (Fig. 11):** We provide a task-by-task breakdown showing specifically where Hierarchical monitoring corrects Sequential failures on complex trajectories.
> * While longer trajectories provide more detection surface area, they require the proposed Hybrid architecture to effectively capitalize on it.
>
> **Conclusion**
> By formalizing the threat model and demonstrating that our Hybrid Monitor generalizes to coding tasks (BashBench) and hard scenarios, we believe we have addressed the core concerns regarding contribution and realism.
>
> ---
> **References**
> [1] Anthropic (2025). Claude 4.5 Sonnet System Card.
> [2] Needham et al. (2025). Large Language Models Often Know When They Are Being Evaluated.
> [3] Greenblatt et al. (2023). AI control: Improving safety despite intentional subversion.
> [4] Korbak et al. (2025). Chain of thought monitorability.
> [5] Panfilov et al. (2025). Capability-Based Scaling Laws for LLM Red-Teaming.

---

> > ### Comment · Reviewer_2pXk · 2025-11-25
> >
> > Thank you for your response, extensive additional experiments, and updated writing. I think this paper should be accepted. I'm raising my score to an 8.

---

> ### Author Response · Authors · 2025-11-25
>
> Thank you very much for your time and for raising your score to an 8. We appreciate your constructive feedback throughout this process. Your questions regarding the limits of CoT monitoring and the necessity of testing generalization were instrumental in improving our paper. We are glad that our revisions and additional experiments satisfied your concerns.

---

### Official Review · Reviewer_gFsT · 2025-10-30

**Soundness:** 4
**Presentation:** 3
**Contribution:** 2
**Rating:** 6
**Confidence:** 4

**Summary:**

The authors examine the effects of various factors on agentic safety monitoring.  Specifically, they explore whether weak models can monitor strong agents; they test how much advantage monitor awareness confers when defending against evasion strategies; they check whether hierarchical monitor scaffoldings are more robust than full-trajectory baselines, and they ask the extent to which human overseers can improve reliability.  Their experiments are performed on the SHADE/CUA-SHADE arena with side tasks for testing computer-use agentic threats.

**Strengths:**

This paper takes a more scientific approach to agentic evaluation than most of those that I've read.  Many papers simply build an agentic workflow or defense system without thoroughly probing it from a scientific standpoint.  This paper includes proper statistical measures of success and asks actionable scientific questions.

This paper includes error bars, which is minor but still very good.

Several of the results are both important and interesting.  For instance, the efficacy of different monitor ``achitectures", such as hierarchical versus sequential monitoring was very interesting.

**Weaknesses:**

The research questions are stated extremely late in the paper (line 252, page 5).  I think it would be better to state these questions as motivations for the research at the beginning of the paper.  These were some of the more interesting questions/findings that you came up with, so it weakens the paper to leave them so late in the draft.

Some of your findings lack depth.  For instance, you check whether Haiku can monitor sonnet, but can you characterize how monitoring capability scales with parameters?  For instance, is the absolute number of parameters in the monitoring agent the most important, or is it the ratio in parameters/capabilities between the attacking agent and the monitoring agent that matters?  Given that this paper is fully empirical, you should seize on opportunities to go deeper into the data analysis.

How general are these findings?  Do they extend to other benchmarks as well?  Why do you rely exclusively on frontier models rather than providing findings on open-source models like Mistral as well?  Can open-source models serve as monitors, given that they're typically even weaker?

**Questions:**

Hybrid monitoring clearly has an advantage over other methods.  However, it is not clear to me what distinguishes the baseline versus sequential and hierarchical methods in terms of efficacy (looking at Figure 5).  Can you comment broadly on when one monitor design is better than another and why?  Do they confer different advantages in different settings?  Where exactly do the advantages of hybrid monitoring come into play?

There's not a clear trend in terms of the chunk size and AUC (appendix D).  How does the AUC decompose for these chunk sizes (i.e. TPR, FPR)?  Is there any trend in those?

Aware attackers use stealthy techniques to avoid detection.  Can you make this more precise by discussing the strategies that they use and the frequency of these strategies?

---

> ### Author Response · Authors · 2025-11-18
>
> We thank the reviewer for the detailed comments. Below we address each concern and describe the corresponding changes (new experiments and analyses).
>
> ### Concern 1: Research questions appear too late.
> > “The research questions are stated extremely late in the paper”
>
> We have **expanded the research questions in Section 1**, explicitly linking them to results to frame the motivation earlier:
> * Agent awareness → **Section 4.2**
> * Evasion strategies → **Section 4.3**
> * Scaffolding designs → **Section 4.2/4.3**
> * Weak-vs-strong → **Section 4.4**
> * Human oversight → **Appendix C**
>
> ---
>
> ### Concern 2: Generality and open-source models.
> > “How general are these findings? ... Why do you rely exclusively on frontier models...?”
>
> We added two open-source monitors (**Mistral-Small-3.1**, **DeepSeek-R1**) against four frontier agents (**Claude Sonnet 3.6**, **Sonnet 4**, **Gemini 2.5 Pro**, **Flash**) on SHADE-Arena (**Section 4.4, Figure 5**).
>
> As shown below, open-source models are effective monitors, particularly with hybrid scaffolding. DeepSeek-R1 even outperforms Claude 3.5 Haiku as a baseline.
>
> **Table: Baseline Monitoring Performance**
> | Agent Model | Claude 3.5 Haiku | Mistral 3.1 Small | DeepSeek-R1 |
> | :--- | :--- | :--- | :--- |
> | **Claude 3.6 Sonnet** | 0.549 | 0.537 | 0.701 |
> | **Claude Sonnet 4** | 0.637 | 0.624 | 0.837 |
> | **Gemini 2.5 Flash** | 0.652 | 0.551 | 0.865 |
> | **Gemini 2.5 Pro** | 0.660 | 0.601 | 0.796 |
>
> **Table: Hybrid Monitoring Performance**
> | Agent Model | Claude 3.5 Haiku | Mistral 3.1 Small | DeepSeek-R1 |
> | :--- | :--- | :--- | :--- |
> | **Claude 3.6 Sonnet** | 0.820 | 0.823 | 0.896 |
> | **Claude Sonnet 4** | 0.877 | 0.886 | 0.884 |
> | **Gemini 2.5 Flash** | 0.848 | 0.853 | 0.899 |
> | **Gemini 2.5 Pro** | 0.798 | 0.840 | 0.875 |
>
> To prove generality, we added **Appendix H** reproducing Findings 1-6 on **BashBench**:
> * **Experimental Setup:** Generated 300+ trajectories (honest, unaware, aware w/ hijacking, lying w/ jailbreaks) using Claude 3.7 Sonnet, Sonnet 4, and Sonnet 4.5.
> * **Evaluation:** Tested 12 monitor configurations (baseline/hierarchical/sequential/hybrid) and 4 monitor models (Sonnet 3.7, Haiku 3.5, GPT-4o, GPT-4o-mini).
> * **Results (Figures 14-16):** The weak-vs-strong scaling curves and hybrid scaffolding advantages replicate fully.
>
> ---
>
> ### Concern 3: Deeper analysis of weak-monitor-strong-agent scaling.
> > “...is the absolute number of parameters... or is it the ratio...?”
>
> We added **Appendix G: Additional Scaling Law Plots**, analyzing performance vs. **Monitor Capability** and **Attacker Capability** (using MMLU/HELM/SWE-Bench as proxies for scale, given that parameter counts for Claude/Gemini are proprietary).
> **Result:** The *ratio* of capabilities is the dominant factor, though the gap is more sensitive to *monitor* capability than *attacker* capability.
>
> ---
>
> ### Concern 4: Hybrid vs baseline/sequential/hierarchical.
> > “Can you comment broadly on when one monitor design is better...?”
>
> We added a failure-mode analysis in **Appendix E** to distinguish these architectures:
> * **Sequential:** Better for shorter trajectories and broad patterns.
> * **Hierarchical:** Better for long trajectories; less vulnerable to recency bias.
> * **Hybrid:** Recovers the benefits of both and is less sensitive to chunk size (**Appendix D**).
>
> ---
>
> ### Concern 5: Chunk size & Evasion strategies.
> > “How does the AUC decompose... (i.e. TPR, FPR)?”
>
> In **Appendix D**, we added:
> * **TPR@FPR=0.01 grid (Figure 7):** Shows that extreme chunk sizes hurt calibration; moderate sizes optimize performance.
> * **Ensemble Baselines (Figure 9):** Confirms hybrid outperforms same-class ensembles (e.g., hierarchical_2 + hierarchical_5), proving gains aren't just from extra compute.
>
> > “...discussing the strategies that they use and the frequency...”
>
> We added **Appendix I** containing:
> * **Table 2:** Full list of evasion strategies (e.g., overwhelm with detail, sound technical).
> * **Quantitative Analysis:** Across 170 trajectories, "Overwhelm with Detail" and "Micro-Steps" were the most frequent (58 trajectories). 6/10 strategies were utilized overall.
>
> **Concluding remarks**
>
> We appreciate the reviewer’s feedback and believe the addition of open-source models (Mistral/DeepSeek), the BashBench generalization experiments, and the deeper scaling analysis significantly strengthen the paper.

---

> > ### Comment · Reviewer_gFsT · 2025-11-24
> >
> > Thank you, the rebuttal is well done.  I'll maintain my score since I still think that the depth of analysis could have been better, but if for some reason your paper looks like it will be rejected I'll argue on your behalf.

---

> ### Author Response · Authors · 2025-11-25
>
> We sincerely thank the reviewer for reviewing our rebuttal and offering to advocate for our paper. In a potential camera-ready version, we will prioritize further expanding the discussion of the scaling laws (Appendix G) and failure modes (Appendix E) to ensure our analysis is as robust as possible. Thank you again for your time and engagement with our work.

---

### Official Review · Reviewer_Nw1E · 2025-11-01

**Soundness:** 2
**Presentation:** 3
**Contribution:** 3
**Rating:** 4
**Confidence:** 4

**Summary:**

This paper introduces a systematic framework for evaluating monitoring systems that detect misbehavior in autonomous LLM agents. The authors propose a structured methodology with formalized threat models (based on agent and monitor awareness levels), standardized evaluation metrics (AUC-ROC, TPR@FPR=0.01), and a taxonomy of adversarial evasion strategies. They contribute CUA-SHADE-Arena, a new benchmark of 15 task pairs in computer-use environments, and evaluate multiple monitor architectures including a novel hybrid approach that combines hierarchical and sequential processing. Key findings include: (1) agent awareness of monitoring substantially degrades detection, while increasing monitor awareness provides limited benefits; (2) a hybrid hierarchical-sequential monitor architecture consistently outperforms baselines; (3) weaker models can potentially monitor stronger agents when using appropriate scaffolding; and (4) structured human oversight can improve TPR by ~15 percentage points at FPR=0.01.

**Strengths:**

The paper makes several contributions, though of varying novelty. The systematic MRT framework provides useful formalization of threat models and evaluation protocols that were previously ad-hoc in the literature. CUA-SHADE-Arena extends monitoring evaluation to more realistic computer-use environments with bash commands and file operations. The hybrid monitoring architecture, while conceptually simple (linear combination of hierarchical and sequential approaches), represents a practical engineering contribution.

The experimental execution is generally thorough with appropriate scale, detailed documentation of prompts and model versions, and reasonable experimental controls.

The paper is well-written with clear structure, helpful figures, and comprehensive appendices.

**Weaknesses:**

**Fundamentally unrealistic threat model**

The core evaluation setup is problematic: agents are explicitly prompted with both tasks and told they're participating in red-team evaluation. As the authors acknowledge in Section 4.1: "we follow SHADE-Arena and frame the agent as assisting in a red-teaming setting." This means:
- Agents are cooperative rather than adversarial
- Misbehavior is prompted, not learned through training/fine-tuning
- "Unaware" agents still know they're in an evaluation setting
- Real threats (goal misgeneralization, emergent deception, adversarial fine-tuning) are not captured

This severely undermines the "weak-to-strong monitoring" claim: showing that Claude Haiku can monitor prompted Claude Sonnet 4 does not demonstrate it could monitor a genuinely superhuman misaligned agent optimized for evasion. The authors mention this limitation (Section 5) but don't adequately address how it affects their conclusions.

Recommendation: Either (a) conduct experiments with adversarially fine-tuned agents, (b) use prompt injection from external sources rather than direct instructions, or (c) substantially temper claims and frame this as preliminary infrastructure work.

**Omniscient monitor performance indicates methodological issues**

If an omniscient monitor knows the exact side task (e.g., "save employee SSNs to .records file"), it should be able to achieve near-perfect detection by simply checking whether that specific outcome occurred. Yet omniscient monitors achieve only 0.90-0.95 AUC for unaware agents and 0.80 AUC for aware agents with evasion (Figure 4).
This suggests the LLM monitors are not performing systematic verification but rather general "suspiciousness" assessment. The prompts (Appendix F) confirm this—they ask monitors to "review each step carefully" rather than "verify if task X was completed."
Implications:

The reported "upper bound" (omniscient performance) is artificially low due to LLM reasoning limitations, not fundamental monitoring limitations
The true performance ceiling might be 1.0, meaning all current approaches are far from optimal
The value of information (comparing unaware/aware/omniscient) is underestimated
Architectural contributions may be overstated—perhaps better prompting would improve all architectures equally

Recommendation: Implement a programmatic omniscient baseline that systematically verifies task completion, report it as the true upper bound, and analyze the gap between programmatic and LLM-based monitoring.


**Insufficient justification for hybrid monitoring**

The paper's main architectural contribution is combining hierarchical and sequential approaches with μ=0.5 linear weighting. However:

- No theoretical explanation: Why are these approaches complementary? What types of attacks does each catch/miss? No failure mode analysis is provided.
- No proper ablations: Missing tests include: (a) comparing to ensemble baselines (e.g., average two hierarchical runs with different chunk sizes), (b) analyzing attack-specific performance to understand complementarity, (c) testing non-linear combinations or learned weighting, (d) examining trajectory length sensitivity.
- Statistical significance unclear: Differences are often small (e.g., 0.838 vs 0.854 in Appendix D Figure 7) with overlapping error bars, yet claimed as meaningful improvements.
- Hyperparameter fairness: Hybrid receives careful tuning (Appendix D) but it's unclear if standalone architectures were equally optimized.
- Conflicting results: Baseline sometimes achieves higher TPR than hybrid (Figure 3, omniscient/unaware condition), contradicting "consistently outperforms" claims.

Recommendation: Add comprehensive ablation studies, failure mode analysis, and statistical significance testing. Either provide theoretical justification or reframe as an empirical engineering finding.

**Questions:**

**Computational costs:**

Can you provide actual measurements of API costs (input/output tokens, dollar amounts) and latency for each architecture? How does hybrid compare to baseline in terms of cost, given that it runs both hierarchical and sequential? Is the 5% AUC improvement worth the additional cost and latency?

**Online monitoring feasibility:**

Your architectures (especially hierarchical and hybrid) require the full trajectory to operate. How do you envision these being adapted for online monitoring where decisions must be made in real-time? What would be the latency per action for sequential monitoring given that it needs ~4 seconds per chunk?

**Time-to-detection:**

For trajectories where malicious actions occur at step N, at what step does each monitor first flag suspicion? What percentage of harmful actions could be prevented if monitoring operated online? How do your offline metrics (AUC, TPR) relate to online effectiveness?

**Action-level vs trajectory-level metrics:**

Your metrics evaluate full trajectories, but online monitoring must evaluate actions. What is the false positive rate per action (not per trajectory)? If monitoring blocks a suspicious action, what's the cost of being wrong (benign action blocked)?

---

> ### Author Response · Authors · 2025-11-18
>
> We thank the reviewer for the constructive feedback. We have conducted additional ablation studies, cost analyses, and text revisions to address the concerns.
>
> ### 1. Threat model realism & “weak-to-strong” claims
> > "Fundamentally unrealistic threat model... Agents are cooperative... Real threats are not captured"*
>
> We agree that our setup (prompted misbehavior + explicit evaluation framing) is a stress-test, not a simulation of emergent AGI deception. We have clarified this scope to avoid over-claiming:
>
> * **Revised Threat Model (Sec 6):** We added a "Threat Model Limitations" section explicitly framing this work as controlled red-teaming infrastructure. We distinguish between risks we approximate (indirect injection, stealth under monitoring) and those out of scope (adversarial fine-tuning).
> * **Tempered Claims:** We adjusted Section 4.4 to refer to “weak-to-strong–style monitoring in a controlled MRT setting,” emphasizing that our results regarding monitor awareness are conditional on this experimental regime.
>
> ### 2. Omniscient monitor and “true upper bound”
> > "Omniscient monitor performance indicates methodological issues... true performance ceiling might be 1.0... reported 'upper bound'... is artificially low"*
>
> You are correct that our "omniscient" baseline measures LLM reasoning limits, not the theoretical ceiling. A programmatic oracle knowing the side task implies AUC=1.0.
>
> * **Clarification:** Rather than adding a static AUC=1.0 line to the plots, we now explicitly distinguish in Section 3.2 between the *Programmatic Oracle* (theoretical ceiling) and the *Omniscient LLM* (upper bound for model-based monitoring).
> * **New Finding:** We added “Finding 3: Omniscient Monitors are Imperfect” (Sec 4.2) to analyze the ‘reasoning gap’ between having perfect information and successfully flagging the trajectory, which varies by awareness setting.
> The omniscient monitor proves better scaffolds can extract sufficient information to approximate "omniscience." If scaffolds failed, defenders would have to learn specific misbehaviors in advance—an unscalable strategy against novel, supercapable agents.
>
> ### 3. Justification and ablations for hybrid monitoring
> > "Insufficient justification for hybrid monitoring... No proper ablations... Statistical significance unclear"*
>
> We have added comprehensive analyses to Appendix D and E to justify the hybrid architecture empirically:
>
> * **Ensemble Baselines (Appendix D, Fig 9):** We tested ensembles (e.g., `hierarchical_2` + `hierarchical_5`) using the same linear weighting. These do not consistently close the gap to hybrid, confirming that hybrid benefits from complementary views (sequential vs. hierarchical) rather than just increased compute or ensembling.
> * **Failure Mode Analysis (Appendix E):** We provide qualitative examples where Hybrid resolves specific failures: Sequential catches long-range patterns missed by Hierarchical, and Hierarchical avoids recency-bias false positives.
> * **Trajectory Length Sensitivity (Appendix E):** We added an analysis showing Sequential performance degrades on longer traces while Hierarchical remains robust, further justifying the combination.
> * **Significance:** We revised our claims to state Hybrid “outperforms or matches” baselines. We explicitly note instances where baselines outperform Hybrid with significance outside the bootstrapped 95% CIs (e.g., low FPR regimes in Fig 3), ensuring we do not claim universal superiority.
>
> ### 4. Computational cost
> > "Can you provide actual measurements of API costs... and latency...?"*
>
> We added **Appendix F: Computational Costs** to quantify the trade-offs:
>
> * **Table 1:** Reports input/output tokens, dollar costs, and latency per trajectory for all architectures.
> * **Analysis:** While Hybrid is more expensive than single-pass baselines, we show it is cost-competitive when using cheaper open-source models (added in Sec 4.4) compared to running larger models in a single pass.
>
> ### 5. Online feasibility & Action-level metrics
> > "Online monitoring feasibility... Action-level vs trajectory-level metrics... What is the false positive rate per action?"*
>
> We agree that online monitoring is the end goal, but our focus here is the offline evaluation of monitor capabilities under MRT.
>
> * **Scope Clarification (Sec 2):** We explicitly frame our contribution as offline trajectory evaluation. Truly online monitoring requires distinct system-level constraints (latency, blocking logic) which we leave for future work.
> * **Contribution:** We provide the necessary prerequisite: a formalized framework and 3,000+ trajectory dataset allows future work to benchmark time-to-detection and online intervention rates.
>
> **Conclusion**
> The revised paper includes a clearer scope, a distinction between reasoning limits and theoretical bounds, and rigorous ablations (ensembles, failure modes) justifying the hybrid architecture. We believe the experiments in Appendices D, E, and F directly address the technical concerns.

---

> > ### Author Response · Authors · 2025-11-25
> >
> > Thank you again for your detailed and constructive review. As the discussion period draws to a close next week, we wanted to gently follow up to ensure you had a chance to review our response above and the new data we generated based on your suggestions.
> >
> > We believe we have addressed your primary concerns regarding the threat model, the omniscient baseline, and the justification for the hybrid architecture:
> >
> > * **Hybrid Architecture Justification (Appendices D & E):** We conducted your requested ablation studies. We found that our hybrid approach outperforms **ensemble baselines** and provided a qualitative failure mode analysis to explain *why* it works.
> > * **Threat Model & Scope:** We revised the text (Section 6) to explicitly frame the work as controlled red-teaming infrastructure rather than a simulation of emergent AGI deception, and tempered our claims regarding "weak-to-strong" monitoring.
> > * **Cost Analysis (Appendix F):** We added the requested breakdown of latency and API costs.
> >
> > We hope these revisions and additional experiments resolve your concerns about the validity of the threat model and the architectural contributions. We would greatly value your feedback on these updates and are happy to answer any remaining questions.

---

### Author Response · Authors · 2025-11-18

We thank all reviewers for their thoughtful and constructive feedback. We are glad that reviewers found our framework well-motivated, thorough, and clearly written, and that they highlighted our research questions and hybrid monitoring results as important contributions to AI control evaluation.

In the revised version, we have:

* Added a **“Threat Model Limitations” subsection** (Section 5).
* Clarified the role of **“omniscient” monitors vs a true programmatic oracle** (Section 4.2, Finding 3).
* Strengthened the **analysis and ablations for hybrid monitoring** (Appendix D, Appendix E).
* Reported **API cost and latency metrics** (Appendix F, Table 1).
* Moved **research questions to the Introduction** and tied them to specific figures (Section 1).
* Added **scaling plots vs absolute agent/monitor capabilities** (Appendix G).
* Included **open-source monitors and a second benchmark (BashBench)** (Appendix H).
* Expanded analyses of **chunk size, evasion strategies, and task difficulty** (Appendix D Figure 9, Appendix E, Appendix I Table 2).

Below we respond to each reviewer in more detail.

---

### Meta-Review · Area_Chair_dW9J · 2026-01-12

**Summary:**

This paper introduces monitor red teaming (MRT) for stress-testing monitoring systems that detect covert misbehavior in LLM-based agents. The authors propose a hybrid hierarchical-sequential monitor design, benchmark it across tool-use and computer-use environments, and demonstrate that weaker models can effectively monitor stronger agents when supported by appropriate scaffolding.
Reviewers praised the paper for its systematic and well-motivated approach to AI safety evaluation, highlighting the formalized threat models, rigorous experimental design with clear metrics, and the introduction of the CUA-SHADE-Arena benchmark. They noted the practical value of the hybrid monitoring architecture, which consistently outperformed baselines and demonstrated a robust weak-to-strong monitoring effect. The work was also commended for being well-written, thorough, and offering actionable insights for real-world AI control.

However, reviewers also raise concerns:

1. Reviewer Nw1E questions (1) The evaluation setup uses prompted misbehavior in a cooperative red-teaming context, which may not reflect real adversarial threats such as fine-tuned deceptive agents. (2) The reported upper-bound performance of omniscient monitors is artificially low due to LLM reasoning limits, not fundamental monitoring constraints. (3)  Insufficient theoretical grounding and ablation studies to explain why the hybrid design outperforms baselines.

2. Reviewer gFsT concerns about  (1) Lack of generalization evidence: Initial experiments relied only on frontier models and a single benchmark. (2) Scaling analysis depth: Needed clearer analysis of whether absolute capability or capability ratio matters most for weak-to-strong monitoring.

3. Reviewer 2pXk questions about (1) the artificial threat assumptions: The premise of situational awareness and prompted malice may not align with real-world risks. (2) Limited evaluation scope: Experiments were confined to one scenario type (AgentDojo-style tasks). (3) Overlap with existing safety intuition: The weak-monitors-strong-agents finding echoes prior “verification is easier than generation” claims, reducing novelty.

4. Reviewer 28fS suggested testing generalization to other coding benchmarks (e.g., BashBench) to strengthen the paper’s contribution.

**Reviewer Concerns:**

For Reviewer Nw1E, the authors addressed several concerns by clarifying the threat model limitations, distinguishing between programmatic and LLM-based omniscient baselines, and adding ablation studies (ensembles, failure modes) for the hybrid monitor. However, the core concern about the realism of the prompted threat model remains outstanding—the evaluation still does not reflect emergent deception or adversarially fine-tuned agents, which limits real-world applicability.

For Reviewer gFsT, most concerns were directly addressed: research questions were moved earlier, open-source models were evaluated, generalization was tested on BashBench, and scaling analyses were added. The rebuttal also provided deeper architectural comparisons and evasion strategy breakdowns. No major concerns remain outstanding.

For Reviewer 2pXk, the authors convincingly addressed concerns about generalization by adding BashBench experiments and clarified the threat model’s scope. They also responded to questions about situational awareness and monitoring access. The concern about novelty relative to “verification is easier than generation” was partially addressed by stressing the architectural contribution, but some may still view the core premise as incremental.

For Reviewer 28fS, the sole request—to test generalization on another coding benchmark—was fully addressed with extensive BashBench experiments in Appendix. All raised points have been satisfactorily resolved.

**Reviewer Scores:**

The authors make a good summary about the reviewers' comments before and after rebuttal.

Reviewer Nw1E: Given the detailed rebuttal that directly addressed their major technical concerns—clarifying the threat model scope, adding ablation studies, and distinguishing omniscient baselines—it is plausible they would have raised their score from a 4 to at least a 6. Their initial critique was rigorous but engaged; seeing the authors’ substantive experimental additions would likely have satisfied many of their methodological reservations.

Reviewer gFsT: They actively participated, maintained their score of 6, and explicitly offered to advocate for the paper. Since they were satisfied with the rebuttal but felt the analysis depth could still improve, their score likely would have remained unchanged. Full discussion might have reinforced their supportive stance without altering their final rating.

Reviewer 2pXk: They fully engaged and, based on the rebuttal, already raised their score from 6 to 8. Further discussion would not have changed this, as they indicated acceptance was warranted and their concerns were addressed to their satisfaction.

Reviewer 28fS: They were highly positive from the start, raised their score from 8 to 10 after the BashBench experiments, and strongly advocated for a spotlight. Full discussion would have solidified their top score, as they viewed the generalized results as a critical contribution.

---

### Decision · Program_Chairs · 2026-01-26

Accept (Oral)